# A Neural Network-Based Random Access Protocol for Crowded Massive MIMO Systems

**DOI:** 10.3390/s23249805

**Published:** 2023-12-13

**Authors:** Felipe Augusto Dutra Bueno, Cézar Fumio Yamamura, Paulo Rogério Scalassara, Taufik Abrão, José Carlos Marinello

**Affiliations:** 1Electrical Engineering Department, Federal University of Technology PR, Av. Alberto Carazzai, 1640, Cornélio Procópio 86300 000, PR, Brazil; felipeaugustodutrabueno@gmail.com (F.A.D.B.); cezaryamamura@gmail.com (C.F.Y.);; 2Electrical Engineering Department, State University of Londrina, Rod. Celso Garcia Cid-PR445, Londrina 86057 970, PR, Brazil; taufik@uel.br

**Keywords:** 6G wireless communication, access protocols, cellular networks, artificial intelligence, energy-efficient wireless systems

## Abstract

Fifth-generation (5G) and beyond networks are expected to serve large numbers of user equipments (UEs). Grant-based random access (RA) protocols are efficient when serving human users, typically with large data volumes to transmit. The strongest user collision resolution (SUCRe) is the first protocol that effectively uses the many antennas at the 5G base station (BS) to improve connectivity performance. In this paper, our proposal involves substituting the retransmission rule of the SUCRe protocol with a neural network (NN) to enhance the identification of the strongest user and resolve collisions in a decentralized manner on the UEs’ side. The proposed NN-based procedure is trained offline, admitting different congestion levels of the system, aiming to obtain a single setup able to operate with different numbers of UEs. The numerical results indicate that our method attains substantial connectivity performance improvements compared to other protocols without requiring additional complexity or overhead. In addition, the proposed approach is robust regarding variations in the number of BS antennas and transmission power while improving energy efficiency by requiring fewer attempts on the RA stage.

## 1. Introduction

Fifth-generation (5G) and beyond (B5G) mobile communication networks should be ready to provide reliable and enhanced mobile broadband (eMBB) communications to an ever-increasing number of devices [1]. The limited availability of time and frequency resources makes pilot collisions prone to happen during the random access (RA) stage, particularly when the number of connected devices exceeds the number of available pilots. This, in turn, can adversely affect the network’s functionality. This problem is an important and challenging issue that future wireless networks must solve to provide reliable connections with the expected quality. Therefore, ensuring the implementation of reliable and effective RA protocols is crucial for the development of B5G networks.

Since its inception [2], massive multiple-input multiple-output (M-MIMO) systems have been evolving from a theoretical concept to a practical technology, becoming a key component of the current 5G standard [3]. Due to user mobility and channel delay spread, the limited channel coherence blocks make the reuse of pilots necessary, which, with a large number of BS antennas, gives rise to a beamformed interference known as pilot contamination [2,4,5,6,7,8,9]. Several different approaches have been proposed with the objective of mitigating this critical impairment of M-MIMO systems, including time-shifted pilots and data transmission between different users [10,11], power allocation [12,13], pilot assignment [13,14], and cell-free (CF)-based schemes [15]. In an RA context, pilot contamination also degrades the performance of crowded M-MIMO systems whenever two or more connected user equipments (UEs) choose the same pilot sequence, which is known as pilot collision [16].

There are multiple possible solutions for pilot collision in M-MIMO systems, such as the adoption of grant-free (GF) or grant-based (GB) protocols. One prominent GB approach is the strongest-user collision resolution (SUCRe) protocol [16], which is a four-step procedure that allows only the strongest contender to access the network resources. The motivation for the strongest-user retransmission criterion is that it will always have only one strongest user in a pilot collision, and it is possible to evaluate a decentralized test to let the user know if it is the strongest contender without additional signaling overhead. Despite being able to address up to 90% of all collisions, the SUCRe protocol still suffers from a significant number of false-negative cases, as highlighted in [17]. This is due to its inability to resolve pilot collisions where the strongest UE’s signal strength is lower than 50% of the sum of the signal strengths of contending UEs.

Some works propose variations on the SUCRe protocol, showing relatively good results. In [18], the UE receives a precoded downlink (DL) response from the BS. This response helps the UEs estimate the sum of the signal strengths of all the competing signals and provides information on idle pilots after the first RA round. Additionally, the response includes an access class barrier (ACB) factor to regulate access control. Hence, some UEs that failed to be granted access on the first attempt can try to access the network resources through previously unused pilot signals. A similar protocol is proposed in [19] in which the BS broadcasts no ACB factor to UEs; instead, a graph-based interference cancellation scheme is applied to maximize the number of UEs that can be admitted to the network. Although both works show better results than the SUCRe protocol, they introduce extra signaling overhead by informing idle pilots to the UEs, which increases latency and harms the system’s spectral efficiency.

Another variation of the SUCRe protocol is the access class barring with power control (ACBPC) RA protocol [20]. The proposed ACBPC protocol suggests implementing decentralized UL pilot transmit power control on the UEs’ end, resulting in a notable performance improvement compared to the SUCRe protocol. Additionally, the ACBPC protocol offers fair access to the UEs, regardless of their distance from the BS. A variation of the SUCRe protocol, known as softSUCRe and introduced in [21], incorporates a soft decision retransmission rule. The softSUCRe rule differs from the SUCRe protocol in that the UE decides to retransmit its pilot based on the probability of being the strongest user. While the softSUCRe protocol yields superior results to the original SUCRe protocol, it also requires additional information for the UE to make decision. Various studies, including [22,23], suggest different approaches to collision resolution protocols. However, Refs. [22,23] introduce additional overhead to the RA phase, increasing system complexity. In addition, other approaches for RA in extra-large MIMO (XL-MIMO) systems are proposed in [24,25].

Finally, the authors of [17] propose a GB RA protocol that employs statistical techniques to address collisions in a decentralized manner at the UE level. This protocol uses a Bayesian classifier (BC) to identify the strongest user and replaces the retransmission rule of the SUCRe protocol. This protocol also shows results superior to the SUCRe protocol without the need for extra overhead. However, as a statistical approach, the BC method has a maximum accuracy limited by the distribution of the considered classes.

With the development of artificial intelligence, a new approach to solving problems in communication systems has emerged, especially using neural networks (NNs) for pattern recognition [26,27]. In this work, we propose a decision-making methodology based on a multilayer perceptron (MLP) NN, which is applied to empirical simulation data of the SUCRe protocol. The proposed approach suggests replacing the retransmission rule of the original SUCRe protocol with an MLP, which can determine if the UE is the strongest contender or not, resolving pilot collisions in a decentralized manner. Our choice for employing offline training instead of online is because the method is evaluated on the UEs’ side since the RA scheme should run in a distributed way for scalability purposes. Thus, the online training should be carried out by the users and would incur additional latency, for they establish a connection to the network, making the approach less attractive. The performance results, which include the fraction of failed access attempts (FFAAs), average number of access attempts (ANAAs), throughput, latency, and energy consumption, indicate a significant improvement in the proposed method compared to other protocols available in the literature.

Thus, the contribution of this paper is threefold:**(i)** We propose a GB RA protocol for crowded M-MIMO systems applying an NN at the UEs’ side to assist in self-classification as the strongest competitor or not, allowing the UEs to resolve pilot collisions in a decentralized and uncoordinated manner;**(ii)** The offline training procedure of the NN to the RA problem is entirely characterized, including data collection, preprocessing, training, and validation steps. In addition, to avoid excessively complex processing at the devices’ side, we show that a simple MLP with only one hidden layer with five neurons is able to remarkably improve the connectivity performance;**(iii)** Extensive numerical results are provided corroborating the performance of the proposed approach, including the performance influence of certain key NN parameters and the robustness against the variation of some network parameters, like the number of BS antennas and transmit power.

The remainder of this paper is organized as follows. The materials and methods are provided in Section 2, which describes the adopted crowded M-MIMO system model in Section 2.1 and proposes the NN-based RA protocol in Section 2.2. In Section 3, the numerical results of the proposed NN GB-RA protocol are presented. The main conclusions are offered in Section 4.

## 2. Materials and Methods

We present in this section our adopted system model for the crowded M-MIMO network under investigation. Then, we present in detail the proposed NN-based RA protocol, carefully describing the methodology for applying the NN to improve the random access performance of the system. In addition, we also evaluate the performance influence of certain key NN parameters, like the number of neurons in the hidden layer and learning rate. Then, in the next section, we numerically evaluate the robustness of the obtained NN regarding the variation of some network parameters, like transmit power and number of BS antennas.

### 2.1. System Model

Similar to the work presented in [16], our M-MIMO system model focus on a center hexagonal cell $C_{0}$, surrounded by 6 neighboring cells $C_{j}$ with j∈{1,2,⋯6}. All cells present a BS located at their centers and equipped with *M* antennas to serve a set of UEs, through a time-division duplex (TDD) scheme, with time and frequency resources divided into coherence blocks of *T* channel uses. Furthermore, we represent the set of all UEs inside cell *j* by $U_{j}$, and the subset of $U_{j}$ of all active UEs by $A_{j}\subset U_{j}$. Also, we consider that inactive UEs will try to become active with probability $P_{a}\leq 1$. Therefore, even in cells with $|U_{j}|\;\gg T$, it is possible to consider a scenario where $|A_{j}|\;<T$. This scenario allows the BS to temporarily make orthogonal payload data pilot (PDP) signals available to all active UEs during payload data transmission by employing a grant-based RA protocol.

Let $K_{0}=U_{0}\setminus A_{0}$ denote the set of inactive UEs with cardinality $K_{0}=\left|K_{0}\right|$ in cell $C_{0}$. The channel vector between BS and UE *k* is denoted by $h_{k}\in C^{M\times 1}$. The channel follows a complex Gaussian distribution $h_{k}∼\mathrm{CN}\left(0,\beta_{k}I_{M}\right)$, where $\beta_{k}$ is the large-scale fading coefficient, obtained as in [16]. The BS makes available a number $\tau_{p}$ of orthogonal RA pilot signals $\left\{\psi_{1},\psi_{2},\cdots \psi_{\tau_{p}}\right\}$, where $\psi_{t}\in C^{\tau_{p}}$ satisfies $\left|\right|\psi_{t}{\left|\right|}^{2}=\tau_{p}$, $t\in \{1,2,\cdots ,\tau_{p}\}$. The available pilots $\tau_{p}$ are then shared by the $K_{0}$ inactive UEs. A particular UE that wants to become active firstly randomly chooses one pilot $\psi_{c(k)}$ out of the $\tau_{p}$ RA pilot signals available and then makes an access attempt by transmitting $\psi_{c(k)}$ with power $\rho_{k}>0$, with $c\left(k\right)\in \left\{1,2,\cdots ,\tau_{p}\right\}$. The UEs choosing the pilot $\psi_{t}$ are represented in the set $S_{t}=\left\{k:c\left(k\right)=t,\;\rho_{k}>0\right\}$, whose cardinality corresponds to the number of UEs contending for such a pilot, and follows a binomial distribution [16]:(1)$$|S_{t}|∼B\left(K_{0},\frac{P_{a}}{\tau_{p}}\right).$$
Given this system model, in order to employ the SUCRe protocol, four steps are necessary, as illustrated in Figure 1.

*(i) Random pilot sequence:* The first step of the SUCRe protocol consists of the UEs sending pilot sequences to the BS, which receives the signal $Y\in C^{M\times \tau_{p}}$ from the sent pilots:(2)$$Y=\sum_{k\in K_{0}}\sqrt{\rho_{k}}h_{k}\psi_{c(k)}^{T}+W+N,$$
where $N\in C^{M\times \tau_{p}}$ is the noise matrix at the BS’s side. Each element of the noise matrix follows $\mathrm{CN}(0,\sigma^{2})$, with $\sigma^{2}$ being the noise variance. $W\in C^{M\times \tau_{p}}$ represents the interference signals received by the BS from the adjacent cells, and ${\left(\cdot \right)}^{T}$ is the transpose operation. The signal Y is then correlated with $\psi_{t}$ at the BS:(3)$$\begin{array}{cc} y_{t} & =Y\frac{\psi_{t}^{*}}{\left|\right|\psi_{t}\left|\right|}=\sum_{i\in S_{t}}\sqrt{\rho_{i}}\left|\right|\psi_{t}\left|\right|h_{i}+W\frac{\psi_{t}^{*}}{\left|\right|\psi_{t}\left|\right|}+n_{t}\\ \; & =\sum_{i\in S_{t}}\sqrt{\rho_{i}\tau_{p}}h_{i}+W\frac{\psi_{t}^{*}}{\left|\right|\psi_{t}\left|\right|}+n_{t}, \end{array}$$
where $n_{t}=N\frac{\psi_{t}^{*}}{\left|\right|\psi_{t}\left|\right|}$ represents the effective noise and has the distribution $\mathrm{CN}(0,\sigma^{2}I_{M})$, with ${\left(\cdot \right)}^{*}$ representing the conjugate operation.

*(ii) Precoded random access:* The second step involves the BS responding to all UEs that transmitted pilot signals by sending a precoded signal $V\in C^{M\times \tau_{p}}$:(4)$$V=\sqrt{q}\sum_{t=1}^{\tau_{p}}\frac{y_{t}^{*}}{\left|\right|y_{t}\left|\right|}\psi_{t}^{T},$$
in which *q* is the BS transmit power per pilot. The *k*-th UE then receives the signal $z_{k}\in C^{\tau_{p}}$:(5)$$z_{k}^{T}=h_{k}^{T}V+\nu_{k}^{T}+\eta_{k}^{T},$$
where $\nu_{k}^{T}\in C^{\tau_{p}}$ is the inter-cell interference (ICI), and $\eta_{k}^{T}$ is noise, which follows $\mathrm{CN}(0,\sigma^{2}I_{\tau_{p}})$. Next, the UE correlates $z_{k}$ with its chosen pilot $\psi_{t}$, resulting in
(6)$$z_{k}=z_{k}^{T}\frac{\psi_{t}^{*}}{\left|\right|\psi_{t}\left|\right|}=\sqrt{q\tau_{p}}h_{k}^{T}\frac{y_{t}^{*}}{\left|\right|y_{t}\left|\right|}+\nu_{k}^{T}\frac{\psi_{t}^{*}}{\left|\right|\psi_{t}\left|\right|}+\eta_{k},$$
where $\eta_{k}∼\mathrm{CN}\left(0,\sigma^{2}\right)$. We define $\alpha_{t}$ as the sum of the signal strengths and inter-cell interference $\omega_{t}$ received by the BS during the first step of the protocol for each pilot *t* in (Equation 3) as
(7)$$\alpha_{t}=\sum_{i\in S_{t}}\rho_{i}\beta_{i}\tau_{p}+\omega_{t}.$$
Then, as proposed in [16], the value of $\alpha_{t}$ can be estimated at the *k*-th UE by ${\hat{\alpha }}_{t,k}$: (8)$${\hat{\alpha }}_{t,k}=\max\left({\left[\frac{\Gamma (M+\frac{1}{2})}{\Gamma (M)}\right]}^{2}\frac{q\rho_{k}\beta_{k}^{2}\tau_{p}^{2}}{{\left[ℜ\left(z_{k}\right)\right]}^{2}}-\sigma^{2},\rho_{k}\beta_{k}\tau_{p}\right),$$
where Γ(·) is the gamma function, and ℜ(·) returns the real part of a complex number.

*(iii) Distributed contention resolution and pilot repetition:* In the third step, it is assumed that the *k*-th UE is aware of its average channel gain $\beta_{k}$. Using this information along with the estimated value ${\hat{\alpha }}_{t,k}$, the UE decides whether to retransmit the pilot signal or not. The primary aim of the SUCRe protocol is to allow only one UE (the strongest one) to retransmit the pilot signal at this step and establish a connection with the network to transmit payload data. Retransmission of the pilot signal occurs when the hypothesis test $R_{k}$ is true [16], and the pilot signal is not retransmitted when $I_{k}$ is true:(9)$$R_{k}:\rho_{k}\beta_{k}\tau_{p}>\frac{{\hat{\alpha }}_{t,k}}{2}+\epsilon_{k},$$
(10)$$I_{k}:\rho_{k}\beta_{k}\tau_{p}\leq \frac{{\hat{\alpha }}_{t,k}}{2}+\epsilon_{k},$$
where $\epsilon_{k}\in R$ is a bias parameter. It is worth noting that (Equation 9) and (Equation 10) mean that the UE only retransmits when its own signal strength is larger than half of the contending UEs’ signal strengths, which is a sufficient but not necessary condition for being the strongest contender [17]. In addition, the bias parameter can be calibrated to adjust the system behavior; for instance, to maximize the average number of resolved collisions, to minimize the occurrence of false positives (or negatives), or to ensure that at most one UE will transmit the pilot in the third step of the SUCRe protocol when $\epsilon_{k}>0$ for all *k* [16]. A suitable value of $\epsilon_{k}$ is proposed as $\epsilon_{k}=\frac{\delta \beta_{k}}{\sqrt{M}}+\frac{\bar{\omega }}{2}$ in [16], where the factor δ multiplies the standard deviations of $\frac{{∥h_{k}∥}^{2}}{M}$ centered around its mean value $\beta_{k}$ and $\bar{\omega }$ is the average UL interference, assumed to be known at the UE [16,17]. It is given by $\bar{\omega }=E\left\{\frac{{∥W\frac{\psi_{t}^{*}}{∥\psi_{t}∥}∥}^{2}}{M}\right\}$, where the expectation is computed with respect to user locations and shadow-fading realizations. The bias term’s influence on performance is also further investigated in [16].

*(iv) Allocation of dedicated data pilots:* In the fourth step, all UEs that successfully retransmitted their pilots (without collision in the third step) are granted access to exclusive network resources to become active and transmit payload data [28].

### 2.2. Neural Network Classifier

One of the most relevant features of artificial NNs is their capability to learn from the input data samples that express the system’s behavior. Hence, after the network has learned the relationship between inputs and outputs in a supervised way, it can generalize solutions, meaning that the network can produce an output close to the expected (or desired) output of any given input value.

This section presents a new approach to tackle the random access problem in crowded M-MIMO networks using an NN classifier specifically designed to resolve pilot collisions under the “strongest-user criterion”. To accomplish this, users are classified into two classes: $Z_{0}$ for those who are not the strongest contenders for their selected pilots, and $Z_{1}$ for the strongest users. The set of classes Z is defined as $Z_{0},Z_{1}$. Each user’s state *k* is denoted as $\Omega_{k}$ and belongs to Z. To approximate the true class of each user, an MLP NN estimates a function $\Omega_{k}=f\left(x_{1},x_{2}\right)$ that maps input values $x_{k}^{\prime }=\rho_{k}\beta_{k}\tau_{p}$ and $x_{k}^{\prime \prime }={\hat{\alpha }}_{t,k}$ to the state $\Omega_{k}$ of the *k*-th UE. In essence, the objective is to obtain an approximation ${\hat{\Omega }}_{k}$ that accurately represents the UE’s true class. Furthermore, as the NN algorithm is evaluated on the devices’ side, we seek here the simplest NN topology able to achieve the desired performance improvements.

The steps to implement this method are the following: *(a) database acquisition*, *(b) preprocessing*, *(c) neural network training*, and *(d) validation*.

#### 2.2.1. Database Acquisition

The first step is to acquire the NN’s training data. The database is generated from the simulation setup publicly shared by the authors of [16], where the values of ${\hat{\alpha }}_{t,k}$, $\beta_{k}$ and their respective actual states $\Omega_{k}$, which serve as labels, are collected.

The numerical parameters for data collection in the SUCRe protocol simulation are shown in Table 1. In this work, we collected nearly $5\times 10^{6}$ labeled training data from the numerical simulation setup with different channel and system scenarios.

#### 2.2.2. Preprocessing

In the preprocessing step, the data are prepared for being used as input for the NN. This step, which includes data shuffling and normalization, is essential for ensuring the proper functioning of an NN. The data shuffling prevents overfitting, and normalization is important to ensure all input data have the same scale and fall within the range of the chosen activation function.

Given the skewed nature (the skewed nature of the data comes from the fact that the dataset has a number of data items labeled $Z_{0}$ much larger than the ones labeled $Z_{1}$) of the collected dataset (primary dataset), the input values are randomly shuffled using the MATLAB function *randperm* to ensure that a subset of the primary dataset with a sufficiently large number of elements will have members of both classes in a proportion near or equal to the one of the primary dataset. Next, $10\times 10^{5}$ data samples are taken from the shuffled dataset, from which 80% are separated for the training set T and 20% for the validation set V. Then, the input values $x_{k}^{\prime }$ and $x_{k}^{\prime \prime }$ of both the training and validation data are normalized according to
(11)$${\bar{x}}_{k}^{\prime }=\frac{\ln\left(x_{k}^{\prime }\right)-\min_{j\in T}\left(\ln\left(x_{j}^{\prime }\right)\right)}{\max_{j\in T}\left(\ln\left(x_{j}^{\prime }\right)\right)-\min_{j\in T}\left(\ln\left(x_{j}^{\prime }\right)\right)},$$
and
(12)$${\bar{x}}_{k}^{\prime \prime }=\frac{\ln\left(x_{k}^{\prime \prime }\right)-\min_{j\in T}\left(\ln\left(x_{j}^{\prime \prime }\right)\right)}{\max_{j\in T}\left(\ln\left(x_{j}^{\prime \prime }\right)\right)-\min_{j\in T}\left(\ln\left(x_{j}^{\prime \prime }\right)\right)}.$$

The normalization procedures mathematically described above are divided into two steps. First, the natural logarithm of the input data is taken to smooth large numerical discrepancies (above six orders of magnitude) among the input data, improving the data resolution. Next, the resulting values are normalized to fit within the closed interval [0,1], matching the output range of the activation function in (Equation 16). Finally, both input values are grouped into a vector ${\bar{x}}_{k}$, where an input bias *b* is also appended:(13)$${\bar{x}}_{k}=\begin{array}{ccc} {}[b & {\bar{x}}_{k}^{\prime } & {\bar{x}}_{k}^{\prime \prime }{]}^{T}. \end{array}$$

#### 2.2.3. Neural Network Training

The training process of an NN consists of applying the required ordinate steps for tuning the synaptic weights and thresholds of its neurons to generalize the solutions produced by its outputs. In the proposed method, the normalized data, ${\bar{x}}_{k}$, and the desired output value $\Omega_{k}$ associated with each training sample are used as training data for an MLP NN with one hidden layer, as illustrated in Figure 2. It is noteworthy that our choice for a simple MLP NN with only one hidden layer is motivated by the fact that such an algorithm is carried out at the devices. Therefore, it is interesting to use the less complex scheme able to achieve the desired performance improvements.

The MLP NN consists of a set of linear combiners, called neurons, that control scalar product operations between input vectors, and sometimes an input bias *b*, with synaptic weights to generate a result by applying a given activation function. The input bias *b* counts as an additional input term for the hidden layer’s neurons. The training is carried out through the well-known backpropagation algorithm [29], which updates the set of weight matrices $W_{1}\in R^{\left(L_{I}+1\right)\times L_{H}}$, the weight matrix between the input layer with $L_{I}$ neurons and the hidden layer with $L_{H}$ neurons, and $W_{2}\in R^{L_{H}\times L_{O}}$, the weight matrix between the hidden layer with length $L_{H}$ neurons and output layer with $L_{O}$ neurons. Given a learning rate κ, the backpropagation algorithm proceeds iteratively by minimizing the mean square error (MSE) function between the desired outputs ${\hat{r}}_{k}$ and the actual output $r_{k}$ at each *i*-th training epoch:(14)$${\mathrm{MSE}}_{i}=\frac{1}{2\;|T|}\sum_{k\in T}{\left(r_{k}-{\hat{r}}_{k}\right)}^{2},$$
where ${\mathrm{MSE}}_{i}$ is the MSE value at the *i*-th training epoch. The training is considered complete when a given precision value in consecutive training epochs, $\xi ={\mathrm{MSE}}_{i}-{\mathrm{MSE}}_{i-1}$ is achieved. Once trained, the MLP NN can be used to estimate the function f(·) as:(15)$$r_{k}=s\left(W_{2}^{T}\cdot s\left(W_{1}^{T}{\bar{x}}_{k}\right)\right),$$
where s(·) is the activation sigmoid function:(16)$$s\left(x\right)=\frac{1}{1+e^{-x}}.$$
Finally, the output $r_{k}$ is associated to one of the output classes in Z. Thus, forming the estimator ${\hat{\Omega }}_{k}$:(17)$${\hat{\Omega }}_{k}=\left\{\begin{array}{cc} Z_{0} & \mathrm{if}\;r_{k}\leq 0.5,\;\\ Z_{1} & \mathrm{if}\;r_{k}>0.5.\; \end{array}\right.$$

#### 2.2.4. Validation

Validation is a crucial step in demonstrating the ability of an NN to generalize its results over data that were not used during the training process. The validation in this work aims to support the choice of the architecture, topology, and the adopted hyperparameter values of the proposed NN binary classifier. The performance metrics *recall*, *precision*, *F-measure*, and *accuracy* are evaluated using the set of data reserved for testing.

The *recall* metric measures the proportion of actual instances of $Z_{1}$ that were correctly classified as $Z_{1}$. *Precision* indicates the ratio of $Z_{1}$ predictions that were actually $Z_{1}$ to the total number of $Z_{1}$ predictions. The *F-measure* is the harmonic mean of precision and recall, calculated as 2×(Precision×Recall)/(Precision+Recall). Finally, *accuracy* represents the proportion of correct predictions made by the MLP NN over all predictions. The results in Table 2 and Table 3 are based on the classification of data generated from a simulation scenario with ICI. These tables show the NN binary classifier performance for different numbers of neurons in the hidden layer $L_{H}$ and different learning rates κ.

In Table 2, the number of neurons in the hidden layer $L_{H}$ increases from 3 to 10 while the learning rate κ is fixed as 0.2 and the precision ξ is set to $10^{-7}$. The bias term is set to b=−1. It is noteworthy from Table 2 that there is no significant change in performance when varying $L_{H}$ from 3 to 10. Also, there is no indication of an increase in the overall accuracy metric. Nonetheless, there is a slight performance increase in the recall and the F-measure metrics. The F-measure is an important metric of performance for skewed data classification since it indicates that the precision and recall metrics are balanced out.

The learning rate in Table 3 ranges from 0.01 to 0.2, with the hyperparameter $L_{H}$ fixed at 5, ξ at $10^{-7}$ and b=−1. As κ increases, the recall drops from 0.7738 to 0.7485, but precision improves slightly from 0.8113 to 0.8355. Despite these changes, the accuracy and F-measure do not present significant changes.

Figure 3 shows the convergence of the MSE function to its minimum value with the training parameters set to $L_{H}=5$, κ=0.2, $\xi =10^{-7}$, and b=−1. The training is set up to stop when one of the two conditions occurs. The first stopping criterion is reached when the required precision value attains $\xi =10^{-7}$, and the second one is completed when a predetermined number of training epochs is attained, which is set to 1000. In Figure 3, the maximum number of epochs is achieved first, which stops the training at 1000 epochs of training. The MSE value achieved is 0.010263, with a precision below $\xi =10^{-6}$.

In the following section, we evaluate the performance of our proposed NN-based RA protocol employing an MLP NN binary classifier with $L_{O}=1$ neuron in the output layer, $L_{I}=2$ neurons in the input layer, and $L_{H}=5$ neurons in the hidden layer. The NN is trained with the backpropagation algorithm with a learning rate κ=0.2. Two main training runs are carried out, the first with data collected in a scenario with ICI and the second with data collected in a scenario without ICI, yielding as the final result of the training process four weight matrices, two of them for the NN trained in the scenario with ICI and two of them for the NN trained in the scenario without ICI. The $L_{H}$ and κ values are chosen empirically among the tested values since their performance metrics do not show any significant difference. The value of $L_{H}$ is also kept relatively low, at 5, because adding more neurons to the NN introduces more complexity, making the algorithm computationally costly. For the same reason, a second hidden layer is not introduced. The bias term, *b*, is set to −1, similar to in [29].

## 3. Results and Discussion

In this section, we report the results of our proposed NN-based RA method applied in an overcrowded M-MIMO scenario, similar to that of [16,17,20,21]. Our outcomes are presented in terms of confusion matrices, ANAAs, FFAAs, latencies, and throughputs. Our study focuses on a system that operates in the 5G sub-6 GHz band, where we consider a center cell named $C_{0}$ with a radius of 250 m [16,17,18,19,20,21,22,24,30]. To create a crowded access scenario, we vary the number of inactive UEs $K_{0}$ in $C_{0}$ from 100 to 30,000 in increments of 500. Although a broad range of $K_{0}$ values is evaluated, our main focus is on the overcrowded scenario because of the very high number of connections expected for B5G networks [1], and since it is a performance bottleneck of the investigated system. Moreover, we add six neighboring cells, $C_{j}$ where j∈{1,2,⋯6}, each having a radius of 250 m and 10 active UEs. Unless specified otherwise, we adopt the parameters from Table 1 in the simulations. We first present the strongest-user classification accuracy of the schemes and then compare the RA performance of the protocols. Our motivation for the strongest-user retransmission criterion is that it will always have only one strongest user in a pilot collision, and our proposed method lets the user itself know if it is the strongest contender in a decentralized way without additional signaling overhead. In addition, since power control is not the focus of this paper, we assume for simplicity the same transmit power for all UEs, similar to in [16,17,18,21,24]. It is worth noting that this is a simple but challenging scenario, since a further elaborated power control mechanism can be employed to improve performance. For example, in [20], a power control mechanism is employed in the RA protocol to provide the same performance for all UEs independent of their distance to the BS at the expense of decreasing the performance of the closest UEs and enhancing the transmit power of the farthest ones to compensate for their severe path loss. Furthermore, we have assumed a transmit power per UE of 27 dBm, which is equal to the BS transmit power per pilot, i.e., ρ=q, as in [16,17,18,19,20,21,22,24].

### 3.1. Classification Performance

Table 4 and Table 5 display the proposed MLP NN-based protocol classification performance. The tables depict the confusion matrices for scenarios with and without ICI. The successful classification rates for each state are shown in the bottom row of the matrix. Additionally, the third column on the far right of the matrix indicates the precision of the predictions for ${\hat{\Omega }}_{k}$. This represents the classifier’s accuracy for each output class ${\hat{\Omega }}_{k}=Z_{0}$ or ${\hat{\Omega }}_{k}=Z_{1}$. The overall accuracy of the classifier is shown in the bottom right square of the matrix.

Table 4 presents the results of the proposed MLP NN-based classifier for a scenario without ICI. The table reveals that the successful classification rate of UEs belonging to class $Z_{0}$ is 99.1%, while the successful classification rate among $Z_{1}$ UEs is 76.5%. It is noteworthy that successful classification rates for $Z_{0}$ are usually higher than for $Z_{1}$. This occurs because when the UE looks at the sum of the contending UEs’ signal strengths and its own signal strength, its decision is usually negatively biased. For example, with SUCRe, the UE only decides positively if its own signal gain is higher than half of the sum of the contending UEs’ signal strengths, which is a sufficient but not necessary condition, leading to high false negative rates. Our proposed NN classifier, on the other hand, improves the classification performance even for the $Z_{1}$ UEs. These rates are higher than those achieved by the SUCRe protocol, 41.7%, and the BC method, 74.8%, as shown in [17]. Moreover, the precision of the $Z_{0}$ and $Z_{1}$ classifications are 98.3% and 85.8%, respectively. The overall accuracy of successful classifications is 97.5%, which is also higher than the values achieved by the SUCRe protocol and the BC method, which are 96% and 97.3%, respectively, in a scenario without ICI [17].

Table 5 presents the results for a scenario with ICI. The reported results indicate that the classification precision is 98.1% for $Z_{0}$ and 84.1% for $Z_{1}$ outputs. Furthermore, in the scenario with ICI, our method achieves a correct prediction rate of 98.9% for class $Z_{0}$ UEs and 74.2% for class $Z_{1}$ UEs. Overall, our method achieves an accuracy of 97.2% in the scenario with ICI, superior to the SUCRe protocol and the BC method.

### 3.2. Connectivity Performance

Figure 4 and Figure 5 depict the numerical results of the ANAAs and FFAAs metrics, respectively, for the following algorithms: **(a)** The baseline, an ALOHA-based protocol described in [16], is represented by a black line and uses a technique where pilot collisions are exclusively addressed by retransmitting them in later RA blocks. **(b)** The results obtained from the original SUCRe protocol, as demonstrated in [16], are represented by red lines marked with “ ∘”. **(c)** The results of the ACBPC protocol [20] are represented by the cyan lines marked with the symbol “▹”. **(d)** The green lines marked with “x” indicate the results obtained with the softSUCRe protocol [21]. **(e)** The outcomes obtained from the BC method presented in [17] are indicated by the magenta lines marked with “⋄”. **(f)** Finally, the blue lines with the “□” marker indicate the MLP NN-based methodology proposed in this paper. The dotted lines show the results obtained for the cases without ICI, while the continuous lines refer to the results with ICI.

The superiority of the proposed method is noteworthy. Comparing the FFAAs results with those of the BC method, for example, the proposed MLP NN-based method achieves better performance when $K_{0}\geq 25,000$ inactive UEs in the cases with ICI, as highlighted in Figure 6, where $K_{0}$ varies from 25,000 up to 40,000 inactive users with probability of activation $P_{a}=0.001$ in a grant-based network operating under $\tau_{p}=10$ pilot sequences. For example, the proposed NN RA protocol reduces the FFAAs ≈5% in comparison with the BC method with 30,000 UEs in the ICI scenario. The proposed NN-based RA protocol also outperforms the BC method in the scenario without ICI, although with a less visible performance gain.

Figure 7 shows the FFAAs results when the number of BS antennas varies from M=1 to M=100 in steps of 2. Even though the NN-based method is trained with M=100 antennas, the proposed method shows robustness under the variation in the number of BS antennas *M*, where a number of M≈50 antennas is revealed to be sufficient to provide superior results than the SUCRe and BC methods in the scenarios with or without ICI.

In Figure 8, we present the performance results with edge SNR variations in dB ($SNR_{dB}$), defined as $\rho \cdot \beta^{e}/\sigma^{2}$ and $q\cdot \beta^{e}/\sigma^{2}$, with $\beta^{e}$ being the large-scale fading of a UE at the cell edge without shadowing and ρ=q, varying from −8 dB to +8 dB. The The FFAAs results are taken for a fixed number of M=100 antennas and $K_{0}=28,000$ UEs. One can see that the performance of the NN-based method, in both scenarios with and without ICI, is superior to the performance of the SUCRe protocol in the whole considered edge SNR range. Compared to the BC method, the proposed NN-based method presents a superior performance in the ICI scenario from ≈−6 dB to ≈+6 dB, and in the cases without ICI, from 0 dB to 8 dB. In addition, although FFAAs results are shown in Figure 7 and Figure 8, it is worth noting that the ANAAs results present a similar behavior, as can be seen from Figure 4 and Figure 5.

In Figure 9, we evaluate the performance of the investigated schemes in terms of average throughput, defined as the number of UEs succeeding in a given RA opportunity divided by the number of RA pilots, $\tau_{p}$. As one can see, the proposed NN RA protocol achieves the highest average throughput in the case without ICI, achieving a throughput of 0.8171 with $K_{0}\approx \tau_{p}/P_{a}=10,000$ UEs. On the other hand, in the case of ICI, the proposed method achieves the best average throughput with a high number of UEs, i.e., with $K_{0}>17,000$ UEs. One can see that the throughput performance of the NN RA protocol in this scenario saturates with ≈0.577, indicating that this ratio of pilot sequences is effectively used in the RA stage independent of the number of UEs. In addition, the proposed method outperforms the state-of-the-art BC protocol in most parts of the $K_{0}$ values.

Figure 10 depicts the average latency performance of the investigated methods, defined as the average time that a UE stays in the RA procedure, i.e., between wanting to become active until succeeding or failing in the RA stage. For this simulation, we have assumed that each RA opportunity takes 1 ms, in the same way as in [24], and that after the first attempt, the UEs decide to try again in each RA opportunity with probability 0.5 [16,17,20,21]. In general, the curves maintain the same shapes as the ones in Figure 4, but scaled by two since, on average, the UEs make an attempt at each two RA opportunities. One can see that the proposed NN RA protocol usually achieves the lowest latencies, allowing the UEs to carry out the RA stage in 12 ms for $K_{0}=3\times \tau_{p}/P_{a}=30,000$ UEs in the ICI scenario.

In addition, to evaluate how the users’ performance depends on their distance to the BS, Figure 11 presents the FFAAs according to the users’ distance to the BS. As one can see, the proposed NN-based RA protocol remarkably improves the performance of the closest UEs, while the performance of the users farther than ≈152 m from the BS becomes nearly the same as the softSUCRe and BC methods and slightly better than the SUCRe performance. On the other hand, the ACBPC protocol of [20] applies UL power control to provide an equal performance for all users, independent of their distance from the BS, and thus provides slightly better performance to the users close to the cell edge. The price to pay for this is significantly decreasing the performance of the closest UEs and increasing the transmit power of the farthest ones to compensate for the severe path loss. It is worth noting that we do not exploit UL power control in the proposed protocol, while assuming a constant UL transmit power for all UEs for simplicity purposes. Nevertheless, if improving the performance of the farthest UEs is an important objective, one can combine a UL power control policy with the proposed protocol, similar to in [20]. However, this is outside the scope of this paper but is suggested as a promising topic for future works.

Finally, it is worth highlighting some insights about the energy consumption during the RA stage. Since the transmit powers are the same for all the investigated schemes, as well as the intervals of each protocol step, we can assume the same energy expenditure per RA attempt for all RA protocols. Therefore, the overall energy expenditure in the RA stage is proportional to the ANAAs, as depicted in Figure 4. One can thus see that the proposed NN RA protocol is the most energy efficient in the scenario without ICI, and in the ICI scenario with $K_{0}>17,000$ UEs. In the latter scenario, the proposed method achieves a ≈24% energy reduction compared with SUCRe, and ≈7% compared to the BC protocol with $K_{0}=3\times \tau_{p}/P_{a}=30,000$ UEs. Consequently, the proposed approach demonstrates itself as a viable GB RA protocol option for B5G systems, outperforming the state-of-the-art BC protocol in the most relevant investigated scenarios.

## 4. Conclusions

In this work, we have proposed a GB RA protocol for crowded M-MIMO systems by implementing an NN on the UEs side to allow their classification under the strongest-user criterion, thus resolving pilot collisions in a non-centralized manner. Based on extensive numerical results, it is possible to conclude that the proposed NN-based method achieves significantly superior performance in comparison with the SUCRe protocol for both scenarios with or without inter-cell interference and without the necessity of extra overhead. We have shown how the proposed method is superior to other state-of-the-art protocols, especially in an overcrowded scenario, i.e., $K_{0}>25,000$ (in scenarios with $P_{a}=0.1\%$ and $\tau_{p}=10$). Finally, we have also evaluated the robustness of our proposed approach concerning the number of BS antennas and transmit power levels, as well as throughput, latency, and energy efficiency performances. Our results demonstrate that our method is a promising solution for implementing a GB RA protocol in crowded B5G systems.

## Figures and Tables

**Figure 1 sensors-23-09805-f001:**
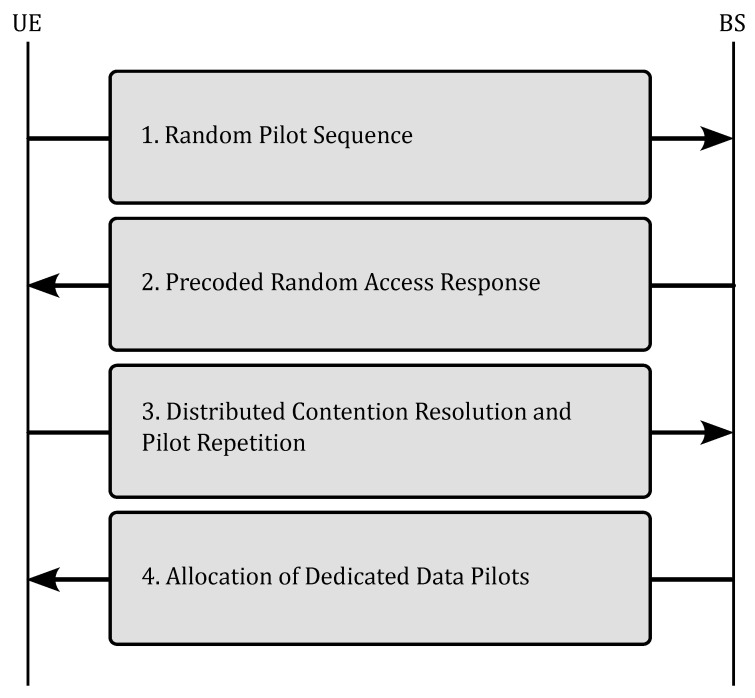
SUCRe protocol diagram for crowded M-MIMO networks.

**Figure 2 sensors-23-09805-f002:**
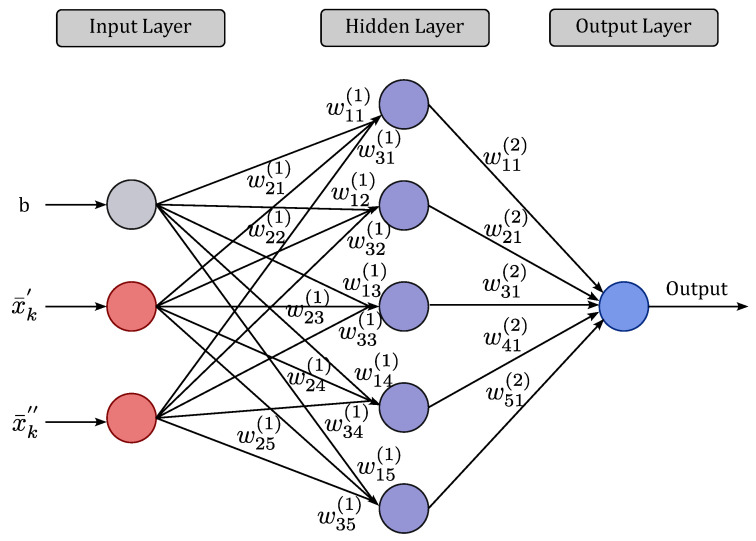
MLP NN with one hidden layer, where $w_{ij}^{l}$ is the synaptic weight between the *i*-th neuron of layer *l* and *j*-th neuron of layer l+1.

**Figure 3 sensors-23-09805-f003:**
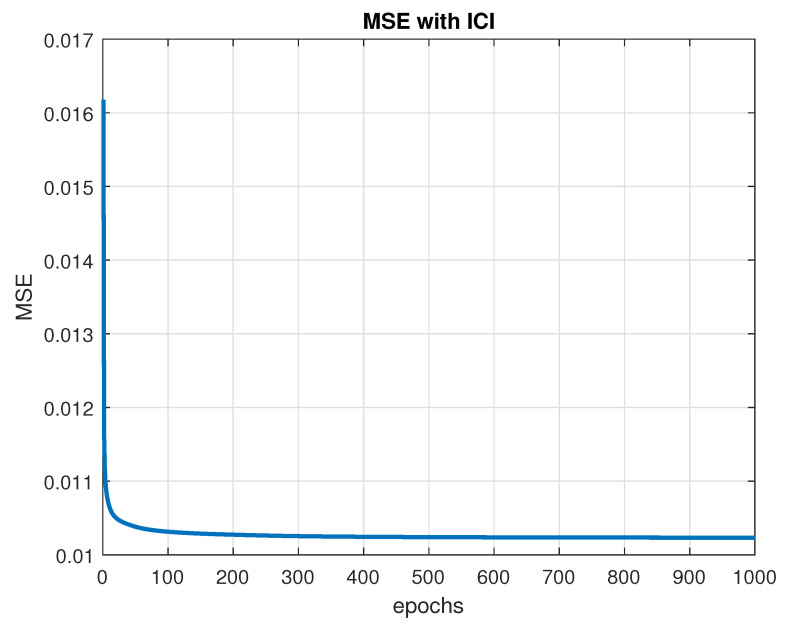
MSE convergence with ICI.

**Figure 4 sensors-23-09805-f004:**
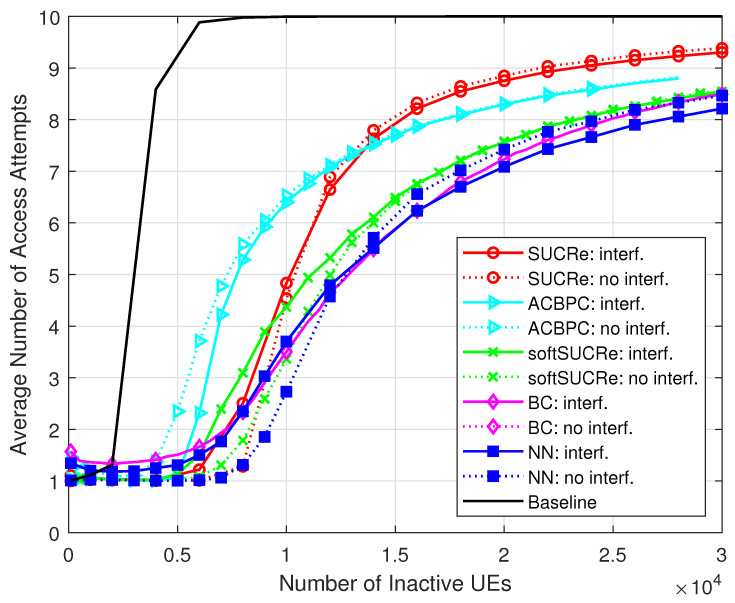
ANAAs $\times K_{0}$, for M=100, $\tau_{p}=10$, $K_{\mathrm{ici}}=10$, and 0 dB of edge signal-to-noise ratio (SNR).

**Figure 5 sensors-23-09805-f005:**
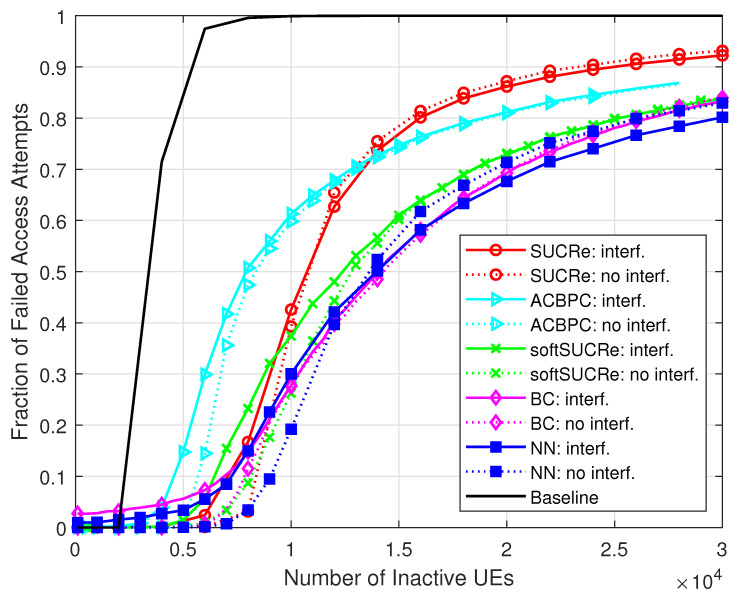
FFAAs $\times K_{0}$, for M=100, $\tau_{p}=10$, $K_{\mathrm{ici}}=10$, and 0 dB of edge SNR.

**Figure 6 sensors-23-09805-f006:**
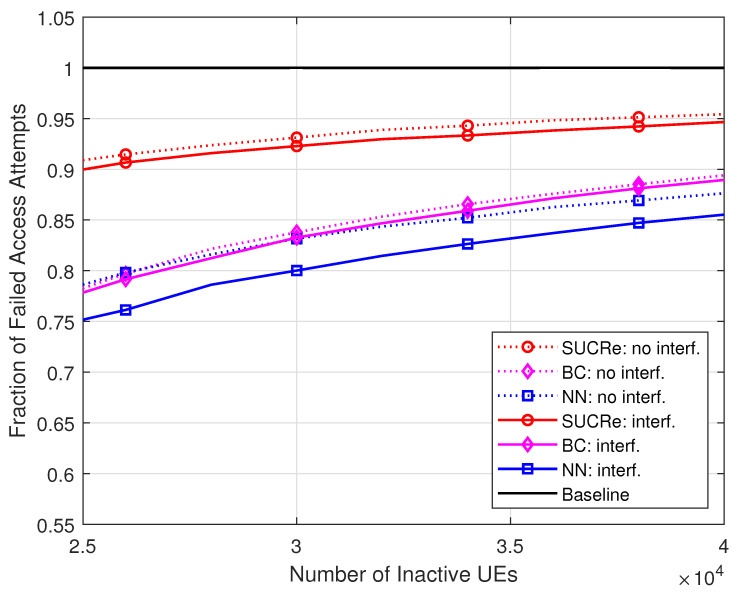
FFAAs $\times K_{0}$, for M=100, $\tau_{p}=10$, $K_{\mathrm{ici}}=10$, 0 dB of edge SNR, and $K_{0}\in \left[25,000,40,000\right]$.

**Figure 7 sensors-23-09805-f007:**
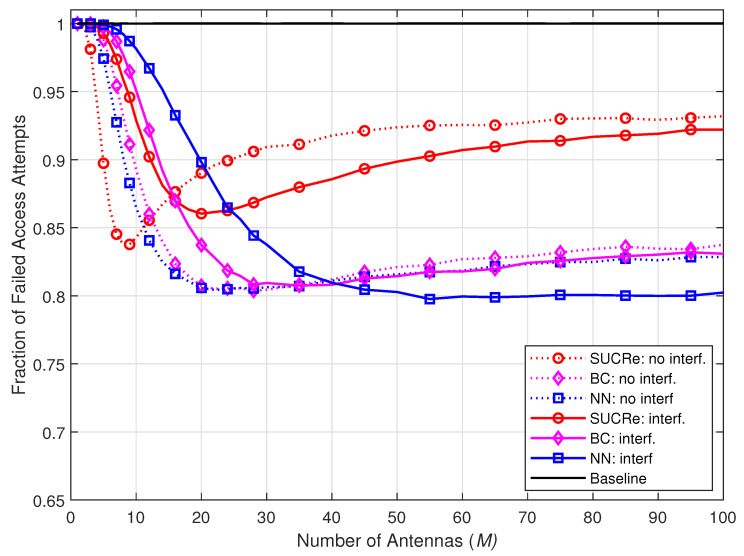
FFAAs performance with *M* variation, considering $\tau_{p}=10$, $K_{\mathrm{ici}}=10$, $K_{0}=28,000$, and 0 dB of edge SNR.

**Figure 8 sensors-23-09805-f008:**
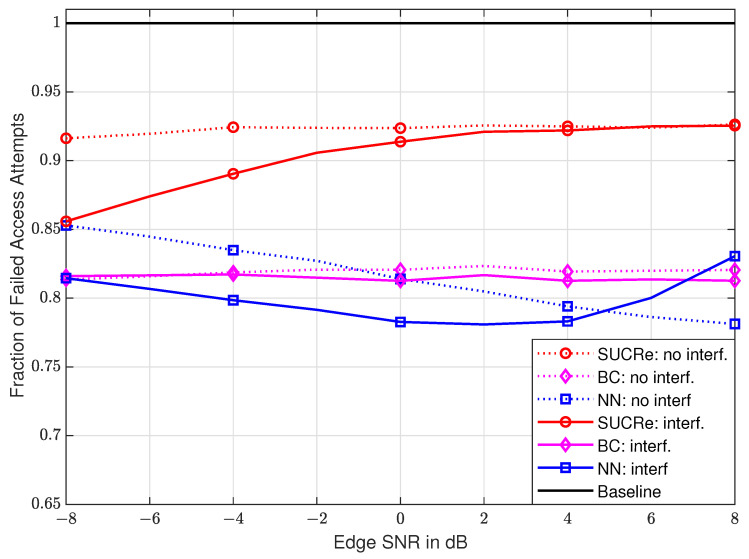
FFAAs performance with edge SNR variation in dB, considering M=100 and $K_{0}=28,000$.

**Figure 9 sensors-23-09805-f009:**
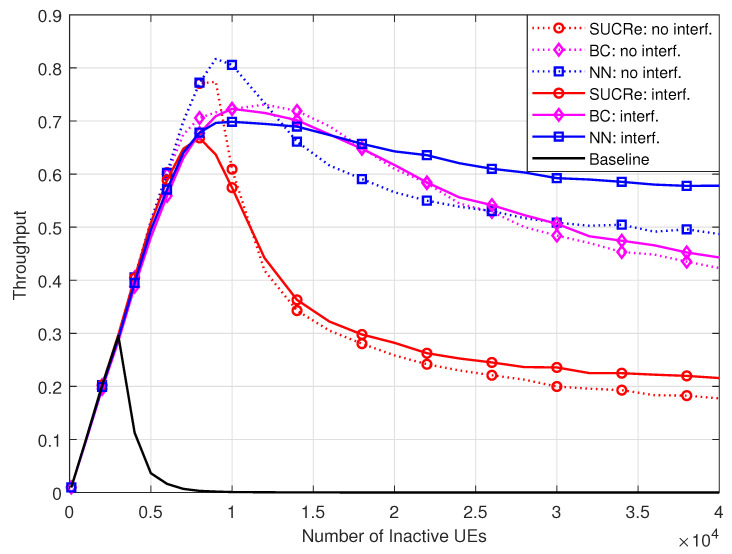
Throughput $\times K_{0}$, for M=100, $\tau_{p}=10$, $K_{\mathrm{ici}}=10$, and 0 dB of edge SNR.

**Figure 10 sensors-23-09805-f010:**
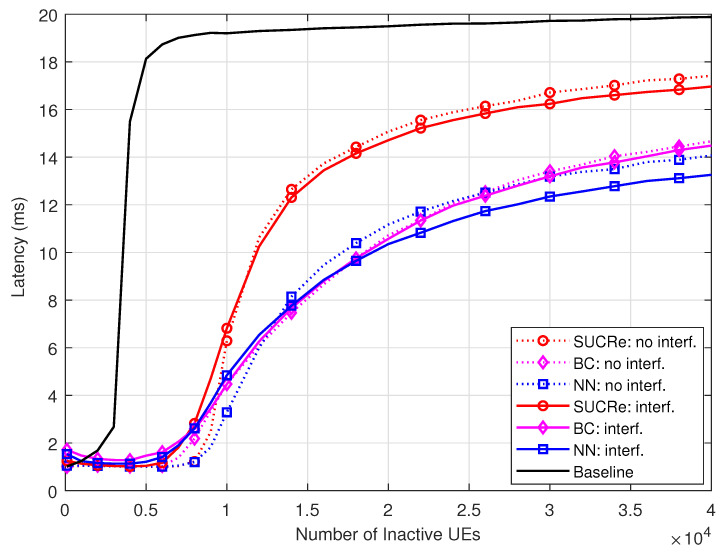
Latency $\times K_{0}$, for M=100, $\tau_{p}=10$, $K_{\mathrm{ici}}=10$, and 0 dB of edge SNR.

**Figure 11 sensors-23-09805-f011:**
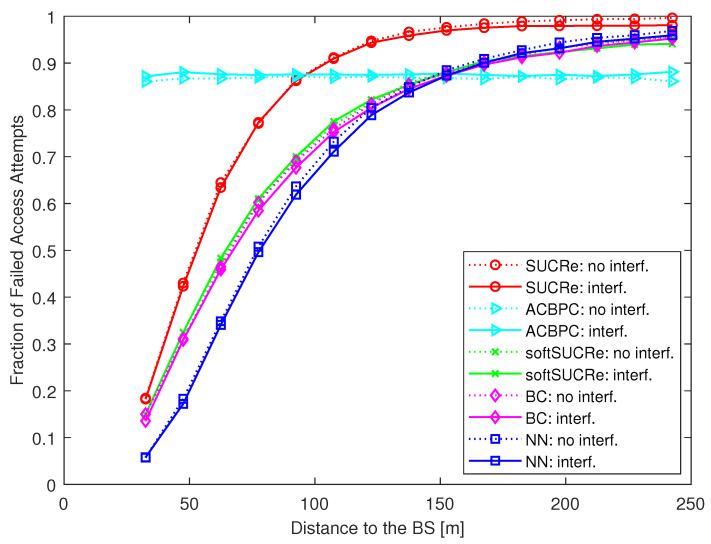
FFAAs according to the users’ distance to the BS, for $K_{0}=28,000$, M=100, $\tau_{p}=10$, $K_{\mathrm{ici}}=10$, and 0 dB of edge SNR.

**Table 1 sensors-23-09805-t001:** Numerical parameters for data collection in SUCRe protocol simulation.

Parameter	Value	Description
*M*	100	Number of BS antennas in the center and neighboring cells
$P_{a}$	0.001	Transmission probability
$P_{r}$	0.5	Probability of trying again in the next RA block
$\tau_{p}$	10	Number of available RA pilot sequences
ρ	27 dBm	Transmit power of the UEs
*q*	27 dBm	Transmit power of the BS per pilot
$\sigma^{2}$	−98.65 dBm	Noise variance
δ	−1	Number of standard deviations in the bias term
$K_{\mathrm{ici}}$	10	Number of active users in each neighboring cell
$K_{0}$	[100, 40,000]	Indicates that $K_{0}\in \left[100,40,000\right]$ in the center cell, evenly distributed in steps of 100 UEs in [100, 1000], and steps of 500 in [1000, 40,000]
Edge SNR	0 dB	Edge SNR in the center cell
	6	Number of neighboring cells
*R*	250 m	Radius of the cells
	27 dBm	Transmit power of UEs in adjacent cells
	10 dB	Shadow-fading standard deviation
	10,000	Number of Monte Carlo realizations
	10	Maximum number of connection attempts before the UE gives up

**Table 2 sensors-23-09805-t002:** Number of neurons test.

$L_{H}$	Recall	Precision	F-Measure	Accuracy
3	0.7414	0.8380	0.7867	0.9719
4	0.7414	0.8380	0.7867	0.9718
5	0.7485	0.8355	0.7896	0.9721
6	0.7567	0.8192	0.7867	0.9713
7	0.7079	0.8532	0.7738	0.9713
8	0.7941	0.7925	0.7933	0.9713
9	0.7763	0.8079	0.7918	0.9718
10	0.7689	0.8081	0.7918	0.9713

**Table 3 sensors-23-09805-t003:** Learning rate test.

κ	Recall	Precision	F-Measure	Accuracy
0.01	0.7738	0.8113	0.7921	0.9718
0.05	0.7556	0.8225	0.7876	0.9716
0.1	0.7280	0.8567	0.7871	0.9720
0.15	0.7617	0.8355	0.7880	0.9714
0.2	0.7485	0.8355	0.7896	0.9721

**Table 4 sensors-23-09805-t004:**
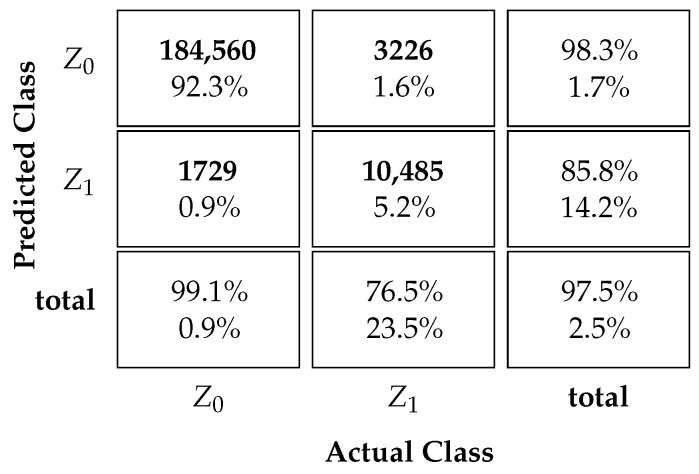
NN classifier without ICI.

**Table 5 sensors-23-09805-t005:**
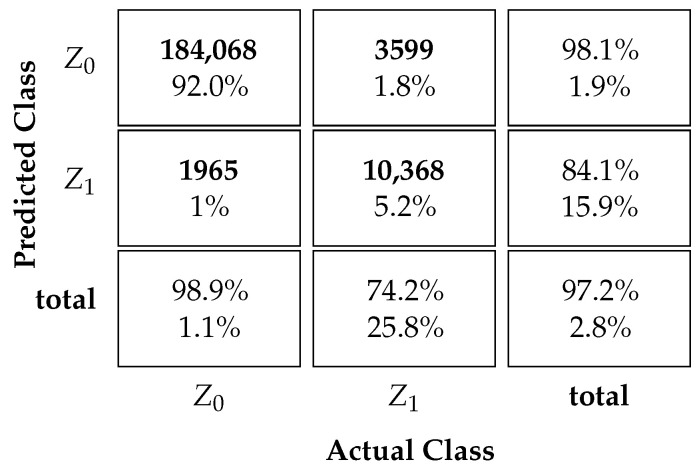
NN classifier with ICI.

## Data Availability

All test data mentioned in this paper will be made available on request to the corresponding author’s email with appropriate justification.

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
