# Peer review of "A Neural Network-Based Random Access Protocol for Crowded Massive MIMO Systems"

_sensors, 2023, doi:10.3390/s23249805_

Round 1

Reviewer 1 Report

Comments and Suggestions for Authors

Author Response

Dear Reviewer,

Thank you for the valuable time and careful reading of our manuscript, as well as for the positive evaluation of our work.

Best regards,
The Authors.

Reviewer 2 Report

Comments and Suggestions for Authors

The use of AI (NN) in actual cellular communication is one of big research doing by many university, this research have some comments as follow:

1. The use of neural network in the methodology and material have to clearly presented as the third contribution is "results are provided corroborating the performance of the proposed approach, including the performance influence of certain key NN parameters, and the robustness against the variation of some network parameters" how the NN method be able to improve random access protocol crowded in massive MIMO system.

2. Figure 2 need to improve the sharpness and figure 1 maybe inner font reduce a bit.

3. The manuscript has a good preparation in term of writing and application for massive MIMO system.

Reviewer 3 Report

Comments and Suggestions for Authors

This paper proposed substituting the retransmission rule of the strongest user collision protocol with a neural network to enhance the identification of the strongest user and resolve collisions in a decentralized manner at the UEs' side. The numerical results indicate that the proposed method attains substantial connectivity performance improvements compared to other protocols without requiring additional complexity or overhead. 

(1) What are the assumptions for open loop power control, power ramping, etc. when doing random access? If we ignore open loop power control, power ramping, etc. won't it be much different from a real system? 

(2) With the proposed random access method, won't UEs with very bad channel conditions consistently fail random access? How do we ensure fairness? To make the proposed random access method meaningful, the authors need to add experiments on fairness. 

(3) This paper seems to consider overly congested environments. Very congested environments can cause performance degradation, so the base station may ask UEs to refrain from random access. Is the proposed method meaningful in a non-congested environment?

(4) Very congested areas will have very small cell sizes. In the numerical parameters, the cell size of 250 meters is not an appropriate parameter. It is necessary to use a smaller value. (For example, 5G system level simulators use the cell size of 66 meters for urban cells.) 

(5) Real-world environments can be very different from experimental environments. If the optimization is done by training offline, can the performance be guaranteed in the real environment? 

(6) If training is possible with a very simple neural network, why not consider online training in addition to offline training? 

(7) In terms of numerical parameters, it is not appropriate for the BS's transmit power to be the same as the UE's transmit power. The BS usually uses 46 dBm or higher, so it may be necessary to change the transmit power of the BS in the simulation. 

Comments on the Quality of English Language

Minor editing of English language required

Reviewer 4 Report

Comments and Suggestions for Authors

The paper deals with the up-to-now well-analysed problem of pilots collision in the LTE systems with multiple antenna systems. The paper is well-written and understandable giving sufficient information for understanding the problem and proposed solutions. The proposed solution slightly outperforms the known BC approach, which limits the interest of the paper to the general public. The results plotted in Figures 4. 5. and 6 are redundant. Both graphs in Fig.7 and Fig.8 actually show the same results but at different scales. I propose selecting only one of the presentations, for example, FFAA. 

Round 2

Reviewer 3 Report

Comments and Suggestions for Authors

The authors responded appropriately to the comments. There are no more comments. 

Comments on the Quality of English Language

Minor editing of English language required.